

# Comprehensive analysis of peroxisome proliferator-activated receptors to predict the drug resistance, immune microenvironment, and prognosis in stomach adenocarcinomas

Qing Jia, Baozhen Li, Xiulian Wang, Yongfen Ma and Gaozhong Li

Department of Gastroenterology, Zibo Central Hospital, Zibo, China

## ABSTRACT

**Background**. Peroxisome proliferator-activated receptors (PPARs) exert multiple functions in the initiation and progression of stomach adenocarcinomas (STAD). This study analyzed the relationship between PPARs and the immune status, molecular mutations, and drug therapy in STAD.

**Methods**. The expression profiles of three PPAR genes (PPARA, PPARD and PPARG) were downloaded from The Cancer Genome Atlas (TCGA) dataset to analyze their expression patterns across pan-cancer. The associations between PPARs and clinico-pathologic features, prognosis, tumor microenvironment, genome mutation and drug sensitivity were also explored. Co-expression between two PPAR genes was calculated using Pearson analysis. Regulatory pathways of PPARs were scored using gene set variation analysis (GSVA) package. Quantitative real-time polymerase chain reaction (qRT-PCR), Western blot, Cell Counting Kit-8 (CCK-8) assay and transwell assay were conducted to analyze the expression and function of the PPAR genes in STAD cell lines (AGS and SGC7901 cells).

**Results**. PPARA, PPARD and PPARG were more abnormally expressed in STAD samples and cell lines when compared to most of 32 type cancers in TCGA. In STAD, the expression of PPARD was higher in Grade 3+4 and male patients, while that of PPARG was higher in patient with Grade 3+4 and age > 60. Patients in high-PPARA expression group tended to have longer survival time. Co-expression analysis revealed 6 genes significantly correlated with the three PPAR genes in STAD. Single-sample GSEA (ssGSEA) showed that the three PPAR genes were enriched in 23 pathways, including MITOTIC_SPINDLE, MYC_TARGETS_V1, E2F_TARGETS and were closely correlated with immune cells, including NK_cells_resting, T_cells_CD4_memory_resting, and macrophages_M0. Immune checkpoint genes (CD274, SIGLEC15) were abnormally expressed between high-PPAR expression and low-PPAR expression groups. TTN, MUC16, FAT2 and ANK3 genes had a high mutation frequency in both high-PPARA/PPARG and low-PPARA/PPARG expression group. Fourteen and two PPARA/PPARD drugs were identified to be able to effectively treat patients in high-PPARA/PPARG and low-PPARA/PPARG expression groups, respectively. We also found that the chemotherapy drug Vinorelbine was positively correlated with the three PPAR genes, showing the potential of Vinorelbine to serve as a treatment drug for STAD. Furthermore, cell experiments demonstrated that PPARG had higher expression

Corresponding author
Gaozhong Li, ligaozhong05@163.com

in AGS and SGC7901 cells, and that inhibiting PPARG suppressed the viability, migration and invasion of AGS and SGC7901 cells.

**Conclusions.** The current results confirmed that the three PPAR genes (PPARA, PPARD and PPARG) affected STAD development through mediating immune microenvironment and genome mutation.

## INTRODUCTION

Gastric cancer (GC) is a malignant digestive system tumor that is most common among patients between 40 and 60 years of age. At present, the mortality of GC is increasing annually, accounting for about 1/4 of all tumor-related deaths worldwide (*Fukayama et al., 2020*). As a malignant pathological phenotype of GC (*Bagaev et al., 2021*), stomach adenocarcinoma (STAD) is largely resulted from unhealthy lifestyle, genetic predisposition and *Helicobacter pylori* infection (*Bagheri et al., 2018*). Currently, STAD is mainly treated by surgery excision, radiotherapy and chemotherapy (*Zeng & Jin, 2022*), but the prognosis of STAD patients remains dismal (*Hoshi, 2020*) due to heterogeneity and metastasis. Though immune and targeted therapies can also improve patients' survival outcomes, we currently face a lack of immune checkpoints and treatment targets specific to STAD (*Zeng & Jin, 2022*). Therefore, discovering potential prognostic markers to facilitate the development of new drugs for STAD is of great significance (*Vyve et al., 2023*).

PPARs, a subfamily of the nuclear hormone receptor family, function as ligand-activated transcription factors to regulate various biological processes. Binding of PPARs to agonist and then to the 9-cis retinoic acid X receptor generates a heterodimer, which then combines with specific peroxide proliferative response element to alter the transcription of target genes and exerts corresponding biological effect (*Chen et al., 2018*). PPAR $\alpha$, PPAR $\beta/\delta$ and PPAR $\gamma$ are the three subtypes of PPARs that differ in tissue distribution, selectivity and sensitivity to ligands. Apart from regulating target gene transcription, the three subtypes of PPARs also regulate various physiological processes including sugar and lipid metabolism and inflammatory response (*Berger & Moller, 2002*; *Monsalve et al., 2013*). *Lu et al., (2005)* found that activation of PPAR $\gamma$ inhibits the occurrence of STAD in mice through apoptosis induced by pioglitazone and tumor suppressor p53. In gastrointestinal tumor cells, PPAR $\gamma$ agonists induce the apoptosis of various STAD cell lines and G1 phase cell cycle arrest to suppress the invasion of GC cells and metastasis (*Chen et al., 2003*; *Sato et al., 2000*; *Takahashi et al., 1999*). A low level of PPAR $\gamma$ in STAD cells enhances fatty acid oxidation (FAO) to promote cancer progression (*Ezzeddini et al., 2021*), and polymorphism of PPAR $\gamma$ is closely associated with the peptic ulcer disease (PUD) and *Helicobacter pylori* infection in STAD (*Prasad et al., 2008*). Another study reported that the CCL20/CCR6 axis mediates PPAR $\delta$ dysregulation to promote STAD carcinogenesis through significantly upregulating

the chemokine Ccl20 level in STAD patients (*Liu et al., 2023b*). These findings suggested the potential of mining PPAR-related genes to improve the current treatment in STAD.

Patients with STAD are often diagnosed at late stages as a result of lack of reliable diagnostic biomarkers and risk factors (*Cover & Peek Jr, 2013*; *Shi et al., 2022*). Advanced next generation sequencing (NGS) facilitates the molecular profiling studies of many cancers, for example, *Chang et al. (2023)* developed a mitochondrial-related gene signature to predict STAD prognosis. The Cancer Genome Atlas (TCGA) repository (https://cancergenome.nih.gov/) is a cancer genome source that stores clinical data, methylation, mRNA expression, gene mutation, miRNA expression, and some other data. Based on the expression pattern of PPARs across pan-cancer in TGCA database, the current study analyzed the signaling pathways and biological functions of the three PPAR genes (PPARA, PPARD and PPARG) to further explore the relationship between the three genes and the prognosis, clinical features, immune microenvironment and traditional chemotherapy drugs in STAD. Our aim was to provide a reliable theoretical basis for the discovery of new potential prognostic markers and therapeutic targets specific to STAD.

## MATERIAL AND METHODS

### Raw data

The RNA-seq and clinical features of 32 type cancers including STAD were downloaded from TCGA dataset. A total of 29 cancer tissue cell lines were collected from The Cancer Cell Line Encyclopedia (CCLE) dataset (*Barretina et al., 2012*).

### Prognosis analysis

Based on the expression of PPAR family genes quantified by a previous study (*Huang et al., 2020*), the STAD samples were classified into low-PPARA/PPARD/PPARG and high-PPARA/PPARD/PPARG expression groups using surv_cutpoint function in survminer package (*Wang et al., 2020*). Differences in overall survival (OS) and progression-free interval (PFI) between two groups were compared using Kaplan–Meier survival curve and log-rank test (*Tan et al., 2023*) in the R package "survival" (*Shen et al., 2023*).

### Co-expression analysis of PPARs and functional enrichment analysis

Co-expression between protein-coding genes and PPARs selected under the cut-off value of |cor|>0.2 and false discovery rate (FDR) <0.05 was calculated by Pearson correlation analysis. Functional annotation of genes (FDR+0.005) to Gene Ontology (GO) items and Kyoto Encyclopedia of Genes and Genomes (KEGG) pathways was performed using clusterProfiler package (*Yu et al., 2012*) and the results were visualized using the ggplot2 package (*Wickham, 2016*).

### Gene set enrichment analysis (GSEA)

Whether a set of genes were statistically significant different between two biological categories could be analyzed using GSEA (https://www.gsea-msigdb.org/). Therefore, we employed GSVA package to perform "c2.cp.kegg.v7.0.symbols" gene set enrichment analysis (*Hänzelmann, Castelo & Guinney, 2013*). Statistically significant genes (*p*-value <0.05) between cancer and normal tissues was assessed by Wilcox.test.
## Tumor immune characteristics associated with the PPARs

In TCGA-STAD dataset, relative abundance of 22 immune cells, such as mast_cells_activated, T_cells_CD4_memory_resting, NK_cells_resting, and macrophages_M0, was calculated by CIBERSORT method. Furthermore, a 29-gene signature representing the main functional components of tumors and some other cell populations was obtained from a previous study (*Bagaev et al., 2021*). Spearman correlation between different immune factors and PPARs were calculated and visualized as heatmaps using the R package "heatmap". The samples were classified into two groups based on the median expression value of the PPAR genes, and the differences of the scores of 22 immune cells between two groups were compared using Wilcox.test.

## Genome mutation

Somatic nucleotide variation of TCGA-STAD samples from the Genomic Data Commons (GDC) dataset were called by Mutect2. Tumor mutation burden (TMB), immune cell proportion score (IPS) and Tumor Immune Dysfunction and Exclusion (TIDE) score were calculated by maftools package (*Mayakonda et al., 2018*).

## Drug response prediction

A total of 26 stomach cell lines treated by 175 drugs were acquired from Genomics of Drug Sensitivity in Cancer (GDSC, https://www.cancerrxgene.org/) (*Yang et al., 2013*). Pearson's correlation analysis was performed to calculate drug sensitivity of cancer cells, and statistically significant PPAR genes were selected under $|cor| > 0.2$ and FDR $< 0.05$. AUC of an antitumor drug served as an indicator of drug response. Half of the maximum inhibitory concentration (IC50) value was calculated by pRRophetic package (*Geeleher, Cox & Huang, 2014*) to reflect drug responses to Docetaxel, Vinorelbine, Paclitaxel, and Cisplatin.

## Cell culture and transient transfection

GES1, AGS and SGC7901 cells were commercially purchased from ATCC (Manassas, USA). Dulbecco's modified Eagle's medium (DEME) was used for cell growth. Negative control (NC) and PPARG siRNA (Invitrogen, Waltham, MA, USA) were transfected into the cells using Lipofectamine 2000 (Invitrogen, Waltham, MA, USA). The target sequence for PPARG siRNA was ACGAAGACATTCCATTCACAAGA (PPARG-si).

## QRT-PCR

Total RNAs of GES-1, AGS and SGC7901 were extracted using TRIzol reagent (Sigma-Aldrich, St. Louis, MO, USA). 500 ng of the RNA were used to synthesize cDNA applying the HiScript II SuperMix (Vazyme, Nanjing City, China). QRT-PCR was carried out in ABI 7500 System using the SYBR Green Master Mix (Thermo Fisher Scientific, Waltham, MA, USA). GAPDH was an internal reference. The sequence list of primer pairs for the target genes was as follow:

## Western blot

Proteins of GES1, AGS and SGC7901 cell lines were lysed in RIPA buffer (Solarbio, China) and denatured at 100 °C for 15 min. Protein samples were isolated using 10% SDS-PAGE

and transferred to polyvinylidene difluoride (PVDF) membranes, which were then blocked by 5% nonfat milk powder solution for 1 h (h). Next, the membranes were incubated with primary antibodies (anti-PPARG antibody (1:1000, 16643-1-AP; Proteintech, Rosemont, IL, USA) and anti-GADPH antibody (1:1000, 60004-1-Ig; Proteintech, Rosemont, IL, USA)) overnight and also with secondary antibodies for 1 h. The signals were visualized using Amersham Hyperfilm electrochemiluminescence (GE Healthcare, Chicago, IL, USA).

### Cell viability

Cell viability of AGS and SGC7901 was measured by performing Cell Counting Kit-8 assay (Beyotime, Jiangsu, China), according to the protocols. The cells treated with or without si PPARG were separately cultured in 96-well plates at a density of $1 \times 10^3$ cells/well. After incubation at 37 °C for 2 h, the OD in each well was measured at 450 nm using a microplate reader.

### Transwell assay

Cell migration and invasion were detected by conducting transwell assay. Briefly, the cells $(5 \times 10^4)$ were added to the chambers coated (for invasion) or uncoated (for migration) with Matrigel (BD Biosciences, Franklin Lakes, NJ, USA). The upper layer and lower layer were added with serum-free medium and full DMEM medium, respectively. After incubation, migrating or invading cells were fixed with 4% paraformaldehyde and stained with 0.1% crystalline violet for 24 h. Finally, the number of cells was counted under a light microscope.

## RESULTS

### Abnormal expression of PPARs across pan-cancer

The expression of PPARA, PPARD and PPARG in a total of 32 types of cancers was determined. We observed that PPARA and PPARG genes were upregulated in most para-cancer samples, while PPARD was upregulated in cancer tissues (Figs. 1A–1C). The expression of PPARG was upregulated in STAD. The three PPAR genes were differentially expressed in 32 primary tumor samples (Fig. 1D) and were dysregulated to varying degrees in 26 out of 29 cell lines, including in GC (Fig. 1E).These results demonstrated that the PPAR genes played important roles in cancer progression.

### The association between the clinicopathologic features and the expression of the PPAR genes

We analyzed the relationship between the expression of PPARs and clinicopathologic staging in GC. There was no statistical difference in the expression of PPARA, PPARD and PPARG genes in Stage I+II and III+IV (Fig. 2A). PPARD and PPARG had higher expression in G3+G4 than that in G1+G2 (Fig. 2B). In samples with an age >60, PPARG was also upregulated (Fig. 2C). The expression of PPARD was higher in males than in females (Fig. 2D). Moreover, STAD samples were divided into high-expression group and low-expression group according to the median expression value of the three PPAR genes. As shown in Table 1, a high PPARD expression was correlated with males. PPARG was correlated with higher grades and older age. Kaplan–Meier curve demonstrated that
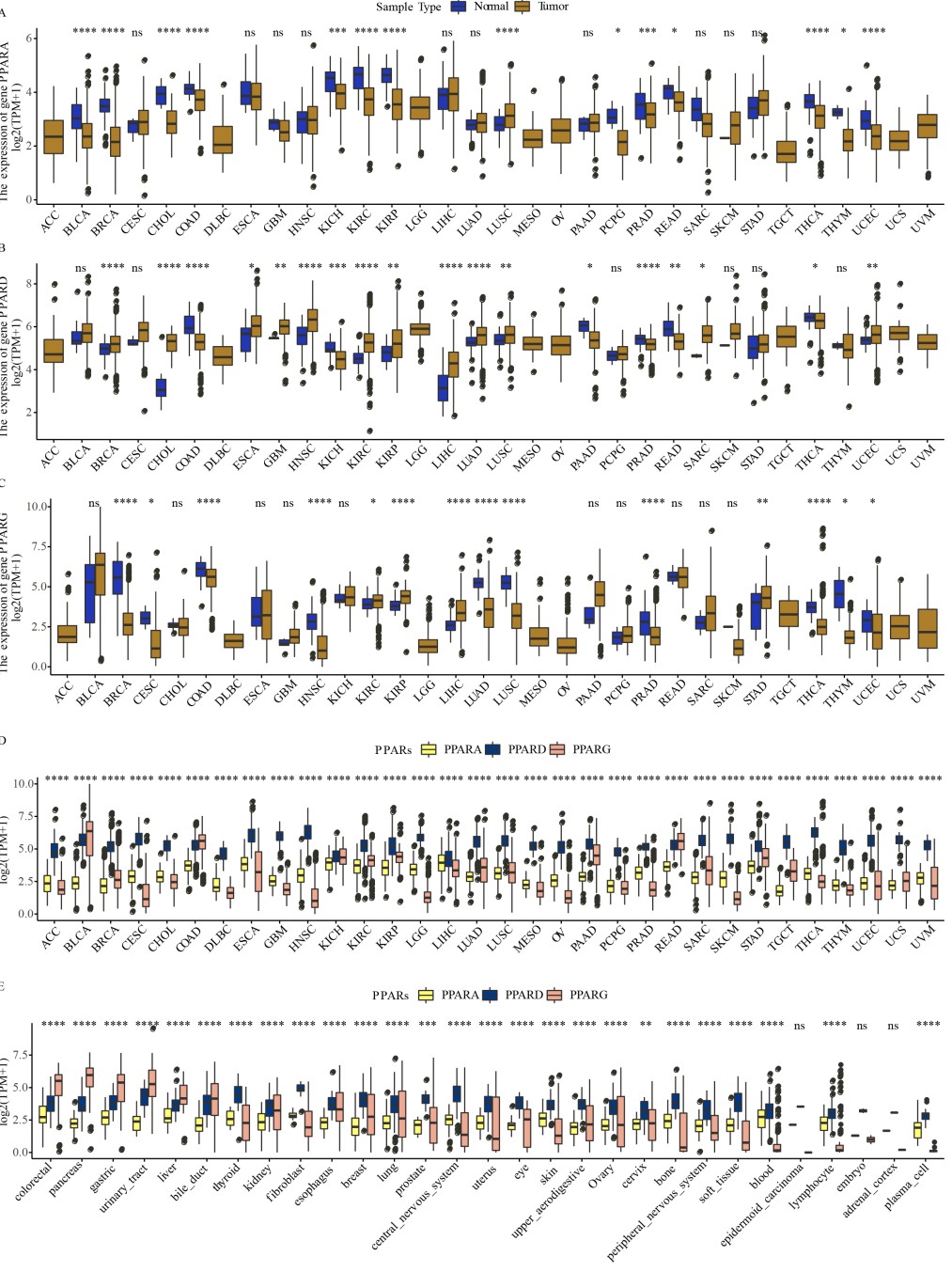

**Figure 1 The expression pattern of PPARs genes in pan-carcinoma.** (A) The expression level of PPARA in 32 type cancers. (B) The expression level of PPARD in 32 type cancers. (C) The expression level of PPARG in 32 type cancers. (D) The expression patterns of PPARs genes in 32 primary tumor samples. (E) The expression of PPARs genes 29 tumor tissue cell lines. (*$p < 0.05$, **$p < 0.01$, ***$p < 0.001$, ****$p < 0.0001$ and "ns" is no significant difference.).

samples in high-PPARA expression group had longer OS and PFI than in the low-PPAR expression group (Figs. 3A, 3B).

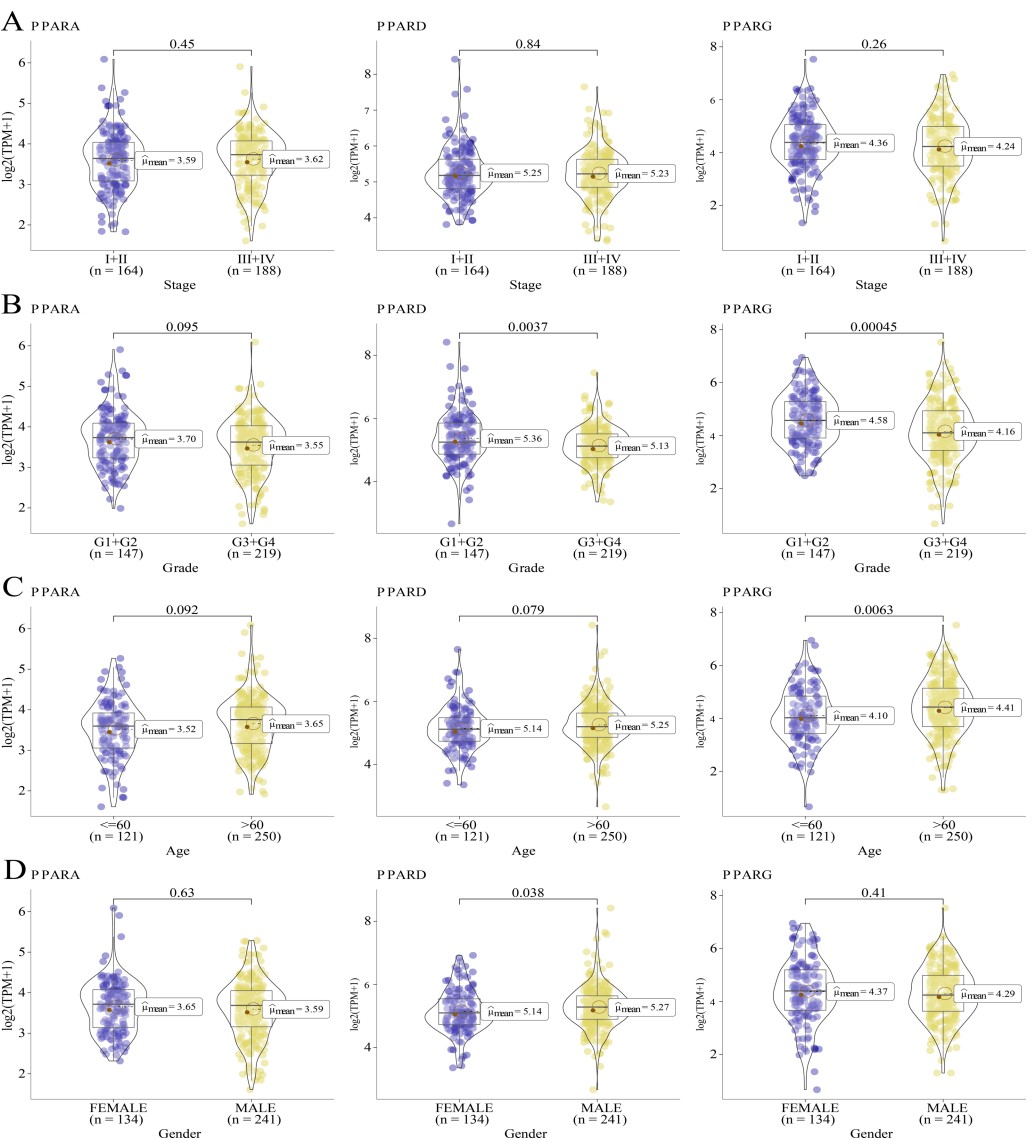

**Figure 2** **The association of the clinical features and PPARs genes.** (A) The expression differences of PPARs genes in Stage I+II and Stage III+IV. (B) The expression differences of PPARs genes in Grade 1+2 and Grade 3+4. (C) The expression differences of PPARs genes in Age<60 and Age>60. (D) The expression differences of PPARs genes in female and male.

## Co-expression analysis of the PPAR genes in STAD

Next, protein-coding genes associated with PPARs were selected by performing Pearson analysis under the criteria of |cor|>0.2 and FDR <0.05. Here, 1,150, 1,244 and 217 genes were correlated with PPARA, PPARD and PPARG, respectively (Fig. 4A), and only six genes (ASAP2, DNM2, EAF1, KIF13B, MFSD9 and TMEM164) were significantly positively correlated with the three PPAR genes (Fig. 4B). In addition, 2,294 genes co-expressing the three PPAR genes were subjected to functional enrichment analysis. The top 20 KEGG pathways were shown in Fig. 4C. Specifically, the 2,294 genes were associated with "process

**Table 1  Relationship between PPARs family member expression and clinicopathological features in the TCGA-STAD cohort.**

| Variable | PPARA | | | PPARD | | | PPARG | | |
|---|---|---|---|---|---|---|---|---|---|
| | High, N = 187[1] | Low, N = 188[1] | p-value[2] | High, N = 187[1] | Low, N = 188[1] | p-value[2] | High, N = 187[1] | Low, N = 188[1] | p-value[2] |
| **T.Stage** | | | 0.90 | | | 0.80 | | | 0.30 |
| T1+T2 | 48 (27%) | 51 (27%) | | 51 (28%) | 48 (26%) | | 53 (29%) | 46 (25%) | |
| T3+T4 | 132 (73%) | 136 (73%) | | 134 (72%) | 134 (74%) | | 127 (71%) | 141 (75%) | |
| Unknown | 7 | 1 | | 2 | 6 | | 7 | 1 | |
| **N.Stage** | | | 0.88 | | | 0.84 | | | 0.76 |
| N0 | 56 (31%) | 55 (31%) | | 56 (31%) | 55 (32%) | | 54 (30%) | 57 (32%) | |
| NX | 122 (69%) | 124 (69%) | | 127 (69%) | 119 (68%) | | 124 (70%) | 122 (68%) | |
| Unknown | 9 | 9 | | 4 | 14 | | 9 | 9 | |
| **M.Stage** | | | 0.56 | | | 0.16 | | | 0.56 |
| M0 | 165 (94%) | 165 (92%) | | 163 (91%) | 167 (95%) | | 165 (92%) | 165 (94%) | |
| M1 | 11 (6.2%) | 14 (7.8%) | | 16 (8.9%) | 9 (5.1%) | | 14 (7.8%) | 11 (6.2%) | |
| Unknown | 11 | 9 | | 8 | 12 | | 8 | 12 | |
| **Stage** | | | 0.33 | | | 0.77 | | | 0.35 |
| I+II | 77 (44%) | 87 (49%) | | 82 (46%) | 82 (47%) | | 85 (49%) | 79 (44%) | |
| III+IV | 98 (56%) | 90 (51%) | | 97 (54%) | 91 (53%) | | 88 (51%) | 100 (56%) | |
| Unknown | 12 | 11 | | 8 | 15 | | 14 | 9 | |
| **Grade** | | | 0.36 | | | 0.12 | | | 0.006 |
| G1+G2 | 77 (43%) | 70 (38%) | | 80 (44%) | 67 (36%) | | 86 (47%) | 61 (33%) | |
| G3+G4 | 104 (57%) | 115 (62%) | | 101 (56%) | 118 (64%) | | 96 (53%) | 123 (67%) | |
| Unknown | 6 | 3 | | 6 | 3 | | 5 | 4 | |
| **Age** | | | 0.046 | | | 0.21 | | | 0.005 |
| <=60 | 51 (28%) | 70 (37%) | | 54 (30%) | 67 (36%) | | 47 (26%) | 74 (39%) | |
| >60 | 133 (72%) | 117 (63%) | | 129 (70%) | 121 (64%) | | 136 (74%) | 114 (61%) | |
| Unknown | 3 | 1 | | 4 | 0 | | 4 | 0 | |
| **Gender** | | | 0.80 | | | 0.006 | | | 0.37 |
| FEMALE | 68 (36%) | 66 (35%) | | 54 (29%) | 80 (43%) | | 71 (38%) | 63 (34%) | |
| MALE | 119 (64%) | 122 (65%) | | 133 (71%) | 108 (57%) | | 116 (62%) | 125 (66%) | |

**Notes.**
[1] Median (IQR) or Frequency (%).
[2] Pearson's Chi-squared test.

utilizing autophagic mechanism," "autophagy," and "histone modification" in BP category (Fig. 4D); the 2,294 genes were related to the "adherens junction", "endosome membrane", and "nuclear speck" (Fig. 4E) in the CC category; the 2,294 genes were related to "GTPase binding" and "cadherin binding" in the MF category (Fig. 4F).

## Pathway enrichment analysis

GSVA package was employed to calculate the pathway score based on hallmark gene set from GSEA website for each patient in the TCGA dataset. As shown in Fig. 5A, 23 pathways were enriched to proliferation-related pathways such as MITOTIC_SPINDLE, MYC_TARGETS_V1, E2F_TARGETS, G2M_CHECKPOINT, MYC_TARGETS_V2 in tumor samples and metabolism-related pathways such as XENOBIOTIC_METABOLISM,

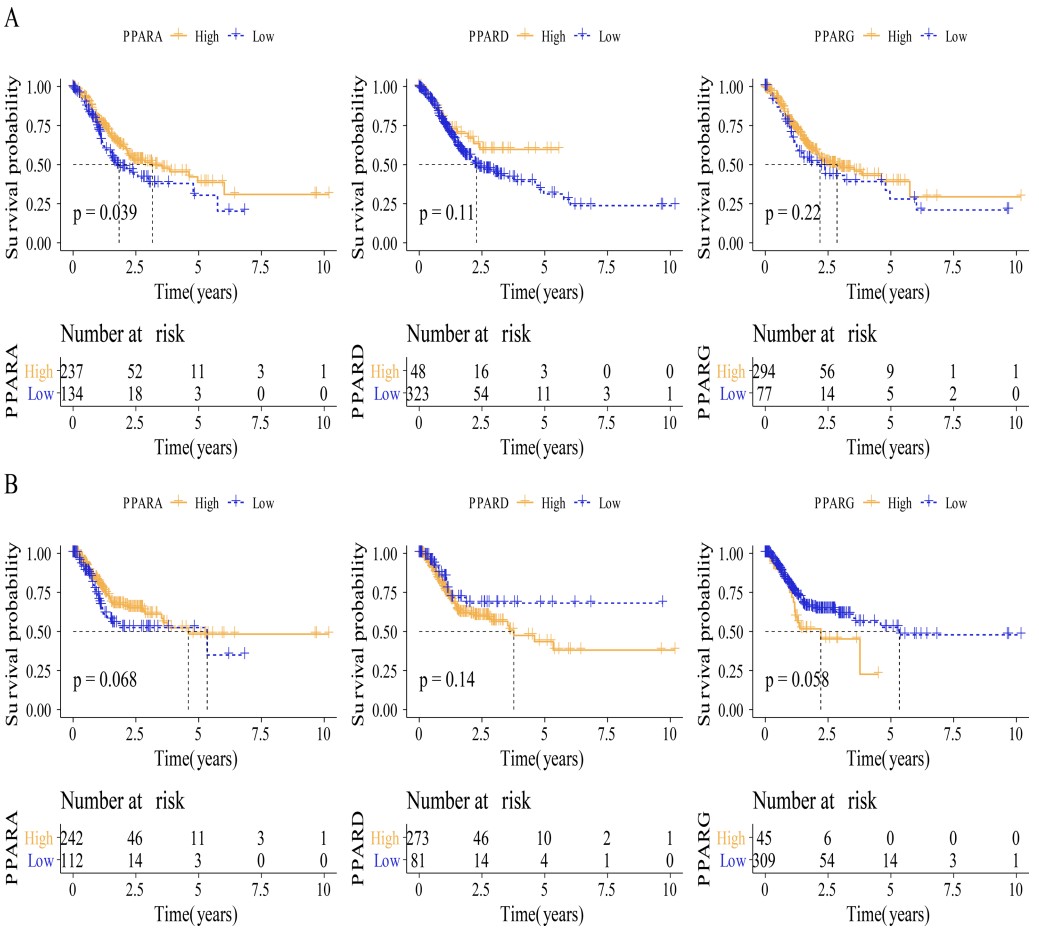

**Figure 3** **The survival analysis between high PPARs genes group and low PPARs genes group.** (A) Patients in high PPARA group had a longer overall survival (OS) than that in low PPARA group. (B) There is no statistical difference of PFI between high PPARs genes group and low PPARs genes group.

BILE_ACID_METABOLISM, FATTY_ACID_METABOLISM, HEME_METABOLISM in normal samples (Fig. 5A). These results indicated that STAD patients could benefit from taking cell cycle-related immune checkpoint inhibitors. Moreover, the association between pathways and the expression of the PPAR genes in both tumor and normal samples was determined. In tumor samples, PPARD and PPARG were correlated with EMT-related pathways and proliferation-related pathways, respectively (Fig. 5B). In normal samples, PPARA was negatively correlated with proliferation-related pathways, and PPARG was positively correlated with immune-related pathways (Fig. 5C). Based on these results, it can be reasonably concluded that these PPARG genes contributed to STAD progression through activating different pathways, but the specific mechanism remained to be further studied.

## Correlation analysis between TME and the PPAR genes

The relative abundance of 22 immune cells was calculated by CIBERSORT analysis, and then the correlation between the PPAR genes and the infiltration of 22 immune

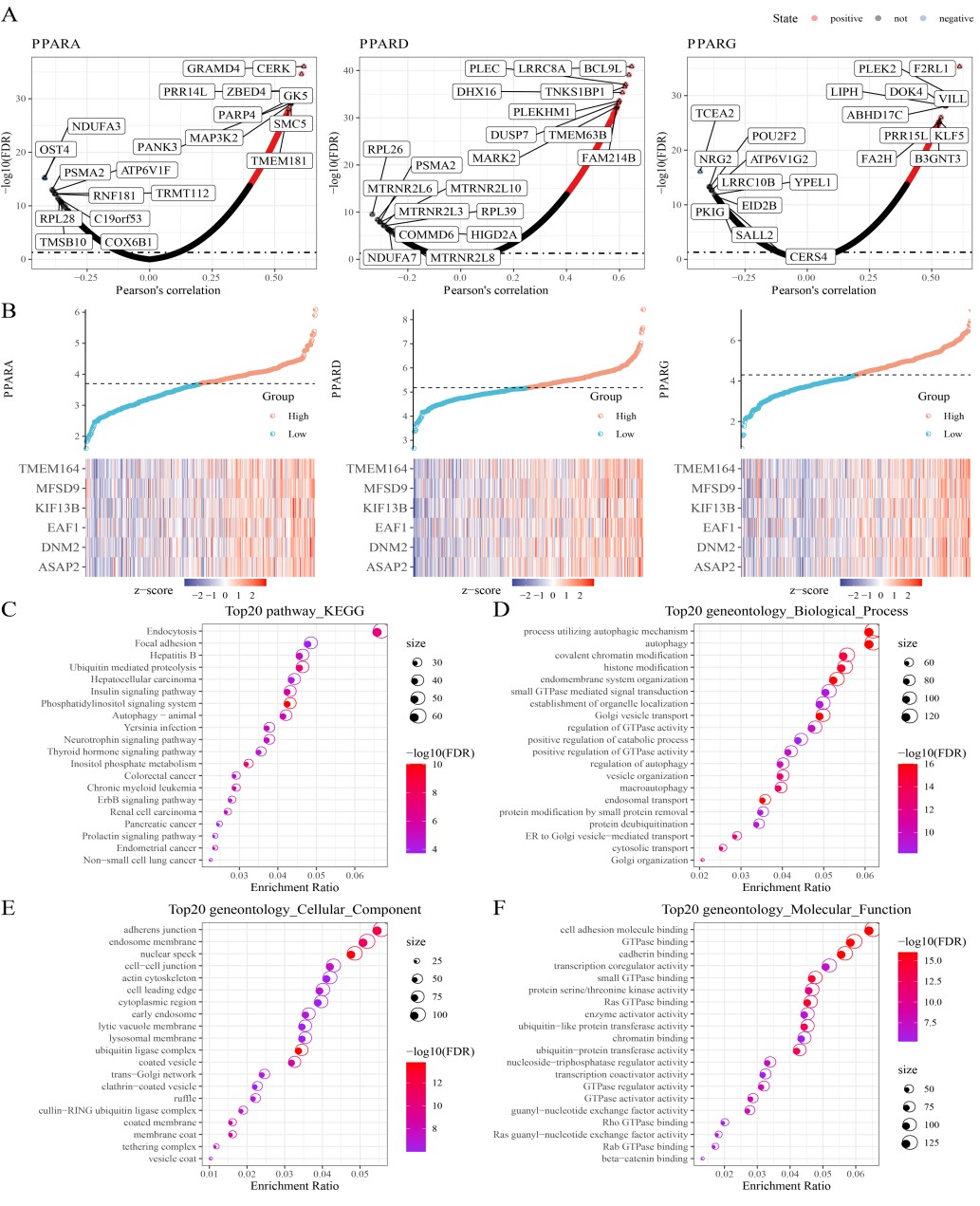

**Figure 4** **Co-expression analysis.** (A) 1,150 genes, 1,244 genes, and 217 genes associated to respectively PPARA, PPARD and PPARG were identified. (B) Six genes (ASAP2, DNM2, EAF1, KIF13B, MFSD9 and TMEM164) both significantly correlated to PPARs genes. (C) The KEGG analysis of 2,294 genes associated to PPARs . (D) GO-biological process analysis of 2,294 genes associated to PPARs. (E) GO-cellular component analysis of 2,294 genes associated to PPARs. (F) GO-Molecular function analysis of 2,294 genes associated to PPARs.

cells was analyzed. We found that PPARA and PPARG were positively correlated with T_cells_CD4_memory_resting, while PPARD was positively associated with

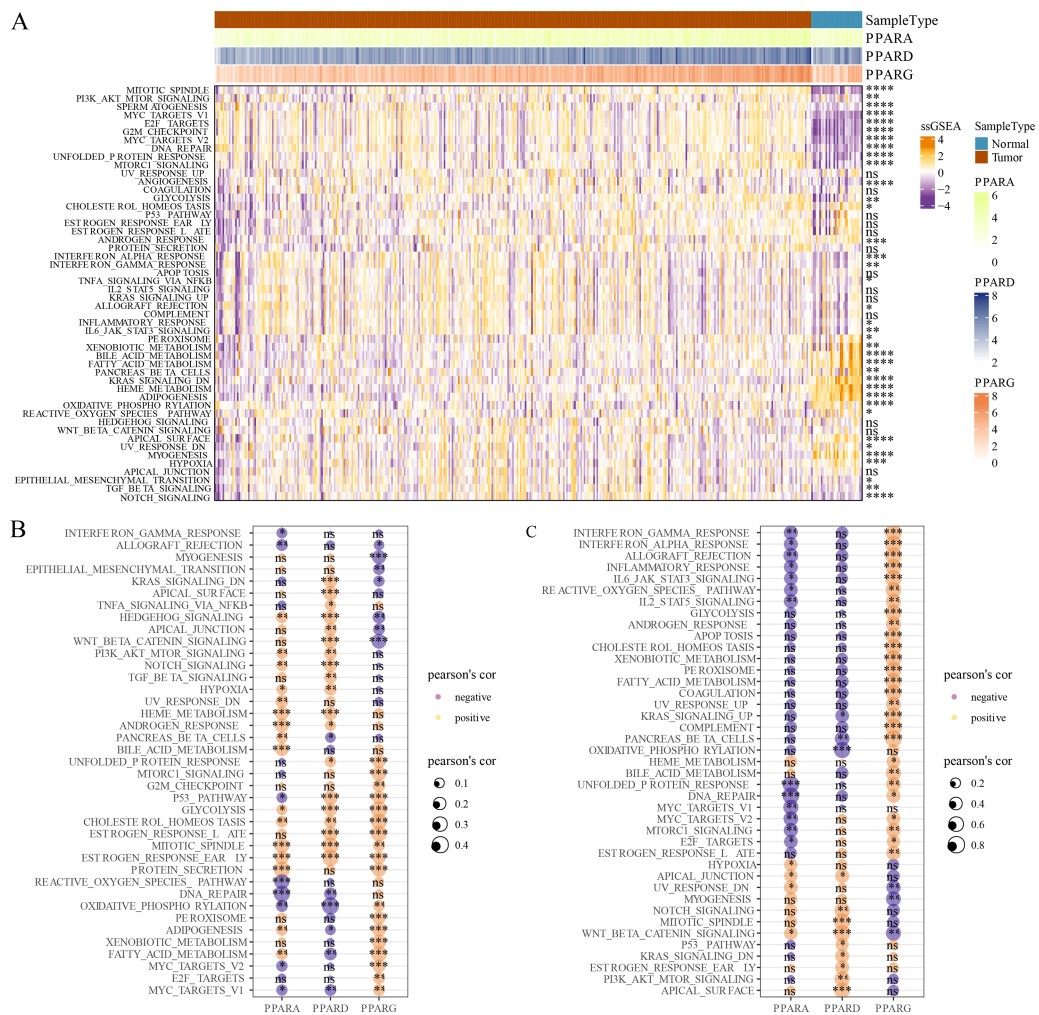

**Figure 5 GSEA analysis.** (A) Twenty-three pathways had significance between cancer tissues and para-cancer tissues. (B) The associated between GSEA pathways and PPARs genes in cancer tissues. (C) The associated between GSEA pathways and PPARs genes in paracancer tissue. (*$p < 0.05$, **$p < 0.01$, ***$p < 0.001$, ****$p < 0.0001$ and "ns" is no significant difference).

NK_cells_resting and macrophages_M0 ($p < 0.05$, Fig. 6A). Immune score of 29 TME-related genes was calculated by ssGSEA. Correlation analysis between the PPAR genes and the TME genes showed that PPARA was significantly negatively correlated with antitumor components, matrix remodeling and tumor proliferation rate, while PPARD was positively correlated with matrix composition ($p < 0.05$, Fig. 6B). Next, we analyzed the relationship between the PPAR genes and seven immune checkpoint genes, and observed that the three PPAR genes were positively related to SLGLEC15 ($p < 0.05$, Fig. 6C). Specifically, PPARD and PPARG had the highest correlation with SLGLEC15 ($R = 0.21$, $p = 3.8e\ 05$) ($R = 0.32$, $p = 4.2e\ 10$), while PPARA had the highest correlation with LAG3 ($R = 0.15$, $p = 0.0039$) (Fig. 6D). Moreover, the expression of the seven immune checkpoint genes in the high-PPAR expression group and low-PPAR expression group was determined.

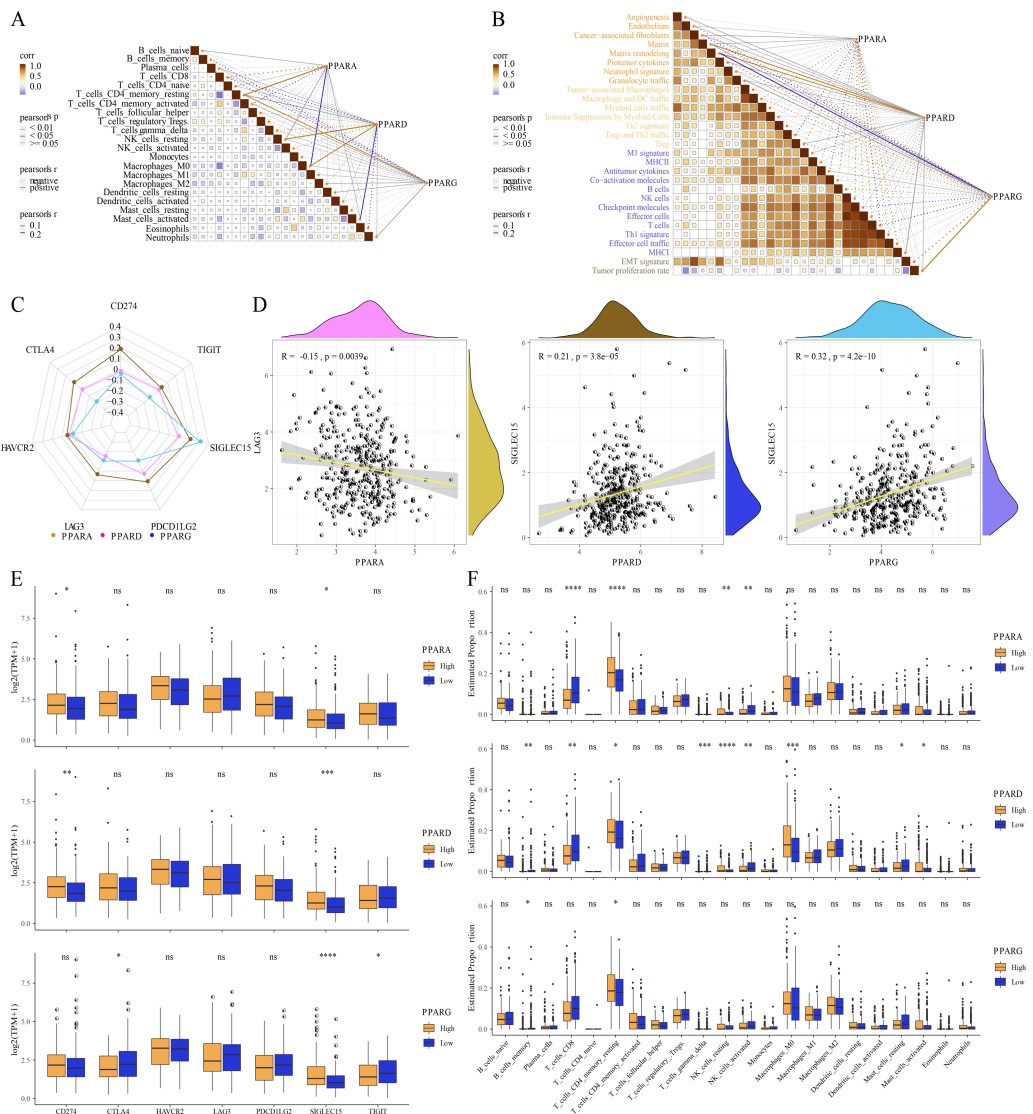

**Figure 6** **The association between Tumor microenvironment and PPARs genes.** (A) The correlation analysis between 22 immune cells and PPARs. (B) The correlation of 29 TME gene signatures and PPARs genes. (C) The correlation between immune checkpoint genes and PPARs genes. (D) The most relevant immune checkpoint associated to PPARs genes. (E) The expression levels of seven immune checkpoint genes between high PPARs expression group and low PPARs expression group. (F) The expression levels of 22 immune cells between high PPARs expression group and low PPARs expression group. (*$p < 0.05$, **$p < 0.01$, ***$p < 0.001$, ****$p < 0.0001$ and "ns" is no significant difference.).

We found that the level of CD274 and SLGLEC15 was upregulated in high-PPARA and high-PPARD expression groups, while CTLA4 and ALGLECL5 were abnormally expressed in high-PPARG and low-PPARG expression groups (Fig. 6E). Finally, the proportion of 22 immune cells in high-PPAR expression group and low-PPAR expression group was calculated, and only few immune cells such as T_cells_CD8, NK_cells_resting showed differences in their proportion between the two groups (Fig. 6F).

## Correlation analysis between genome mutation and the PPAR genes

Analysis on the relationship between genome mutation and the PPAR genes showed that the TMB was positively correlated with PPARA and PPARG (Fig. 7A). At the same time, TMB was higher in the high-PPARA and high-PPARG expression groups than the corresponding low expression groups of the two genes (Fig. 7B). In addition, the immune cell proportion score (IPS) and Tumor Immune Dysfunction and Exclusion (TIDE) score revealed that the patients with high expression of PPARG and PPARA were more likely to benefit from taking immune treatment (Figure S1). TTN, MUC16, FAT2 and ANK3 showed higher mutation frequency in high-PPARA- and high-PPARG expression groups, and DNAH11, HERC2, NIPBL had higher mutation frequency in the high-PPARD expression group (Figs. 7C–7E).

## Drug sensitivity analysis of PPARs

A total of 26 drug-treated STAD cell lines were obtained from GDSC. Pearson analysis was used to calculate the correlation between drug sensitivity and the expression of the three PPAR genes. We identified 14 PPARA-drug sensitivity pairs (Fig. 8A), and there were 8 drug resistance-related pairs and 2 PPARD-drug sensitivity pairs (Fig. 8B). Specifically, OF-1 targeted BRPF2, OSI-027 targeted MTORC2, LGK974 targeted WNT signaling, EPZ004777 targeted chromatin histone methylation, and GI-6780A targeted metabolism (Fig. 8C). Moreover, analysis on the relationship between IC50 of chemotherapy drugs (Docetaxel, Vinorelbine, Paclitaxel and Cisplatin) and the PPAR genes demonstrated that the PPAR genes were positively associated with Vinorelbine (Fig. 8D).

## PPARG promoted the invasion and migration ability of GC cell lines

The results of PCR demonstrated that the mRNA level of PPARG was elevated in the two GC cell lines (AGS and SGC7901) compared to normal gastric epithelial cells GES1 (Figs. 9A, 9B). Consistently, the results of Western blot also showed that the protein level of PPARG was significantly higher in AGS and SGC7901 cell lines (Fig. 9C, 9D). Hence, we hypothesized that PPARG could promote the activity of GC cell lines *in vitro*. The viability of AGS (Fig. 9E) and SGC7901 (Fig. 9F) was significantly lower after suppressing PPARG expression. Transwell assay was performed to detect the migration and invasion ability of AGS and SGC7901 cells after knocking down PPARG. It was observed that the invasion ability of the two GC cell lines was significantly reduced after PPARG inhibition (Figs. 9G–9L). These data indicated that PPARG could significantly promote the viability of GC cell lines.

## DISCUSSION

PAR $\alpha$, PPAR $\beta$ and PPAR $\gamma$ are three subtypes of PPARs encoded by independent genes. Previous studies reported that these three genes are closely related to the occurrence and progression of GC. The mRNA and protein expression of PPAR $\gamma$ is significantly higher in GC than that in normal and paracancer gastric mucosa. Previous studies found that the mRNA and protein expression of PPAR $\gamma$ silenced by mRNAi in human GC MGC803 cells could inhibit cell proliferation (*Ma, Yu & Huai, 2009*). *Lu et al. (2005)* confirmed that

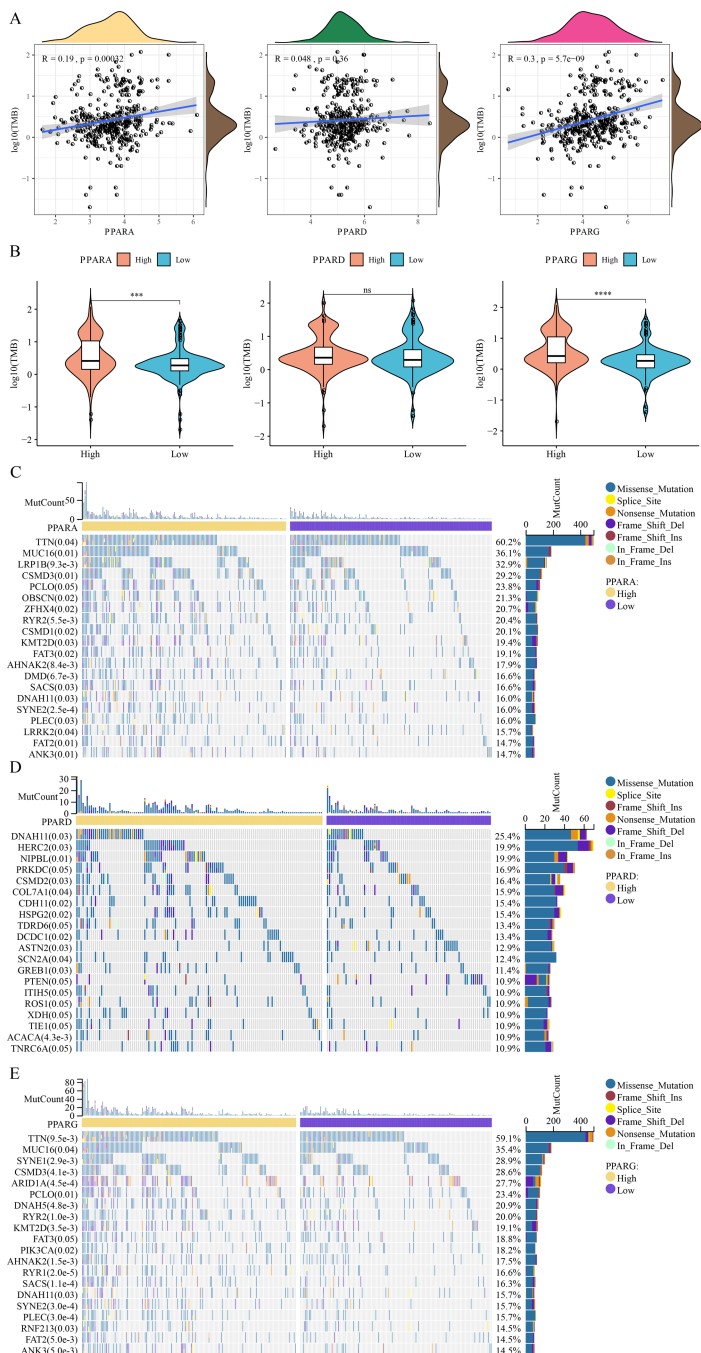

**Figure 7  The association between genome mutation and PPARs.** (A) The correlation analysis between TMB and PPARs genes. (B) The difference of TMB in the high PPARs expression group and the low PPARs expression group. (C) Gene mutation difference between the high PPARA expression group and the low PPARA expression group. (D) Gene mutation difference between the high PPARD expression group and the low PPARD expression group. (E) Gene mutation difference between the high PPARG expression group and the low PPARG expression group. ($*p < 0.05$, $**p < 0.01$, $***p < 0.001$, $****p < 0.0001$ and "ns" is no significant difference.).

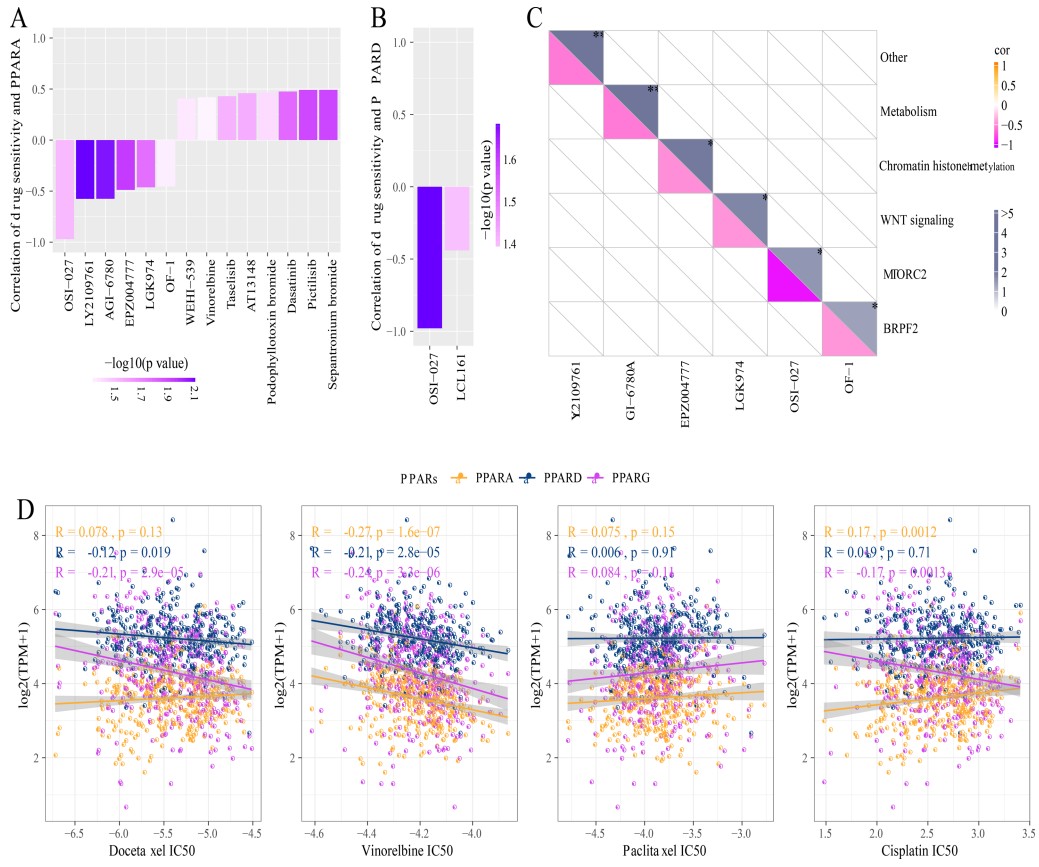

**Figure 8 Drug sensitivity analysis correlated to PPARs genes.** (A) Correlation between PPARA expression and drug response in gastric cancer cell lines in GDSC. (B) Correlation between PPARD expression and drug response in gastric cancer cell lines in GDSC. (C) Drug targeting pathways in GDSC. (D) The box plots of the estimated IC50 for Docetaxel, Vinorelbine, Paclitaxel and Cisplatinin TCGA.

PPAR $\gamma$ ligand troglitazone could reduce the risk of GC induced by chemical carcinogens in rats. In patient-derived GC cells, PPAR $\delta$ knockout significantly suppresses the cancer cell invasion and tumor formation (*Song et al., 2020*). PPAR $\alpha$/$\gamma$ agonist TZD18 inhibits the growth of GC cells by inducing apoptosis (*Ma et al., 2019*). PPAR $\alpha$ shows a high expression in GC and is negatively correlated with prognosis (*Chen et al., 2020b*). At present, there is no systematic analysis probing into the specific effects of the 3 PPAR genes on GC. To bridge such a gap, this study analyzed the three genes across pan-cancer including GC using bioinformatics analyses, and found that the expression of the three genes was dysregulated to different degrees. We also observed that GC patients with high-expressed PPARA had better prognostic outcomes, while those with high-expressed PPARG exhibited immune regulation potential to control STAD progress. This suggested that the PPAR genes played important roles in the progression of GC.

Co-expression analysis identified six genes that were closely associated with the three PPAR genes. ASAP2, DNM2 and KIF13B encoding signal transduction proteins play crucial roles in tumor proliferation and invasion. *Chen et al. (2020a)* showed that high-expressed

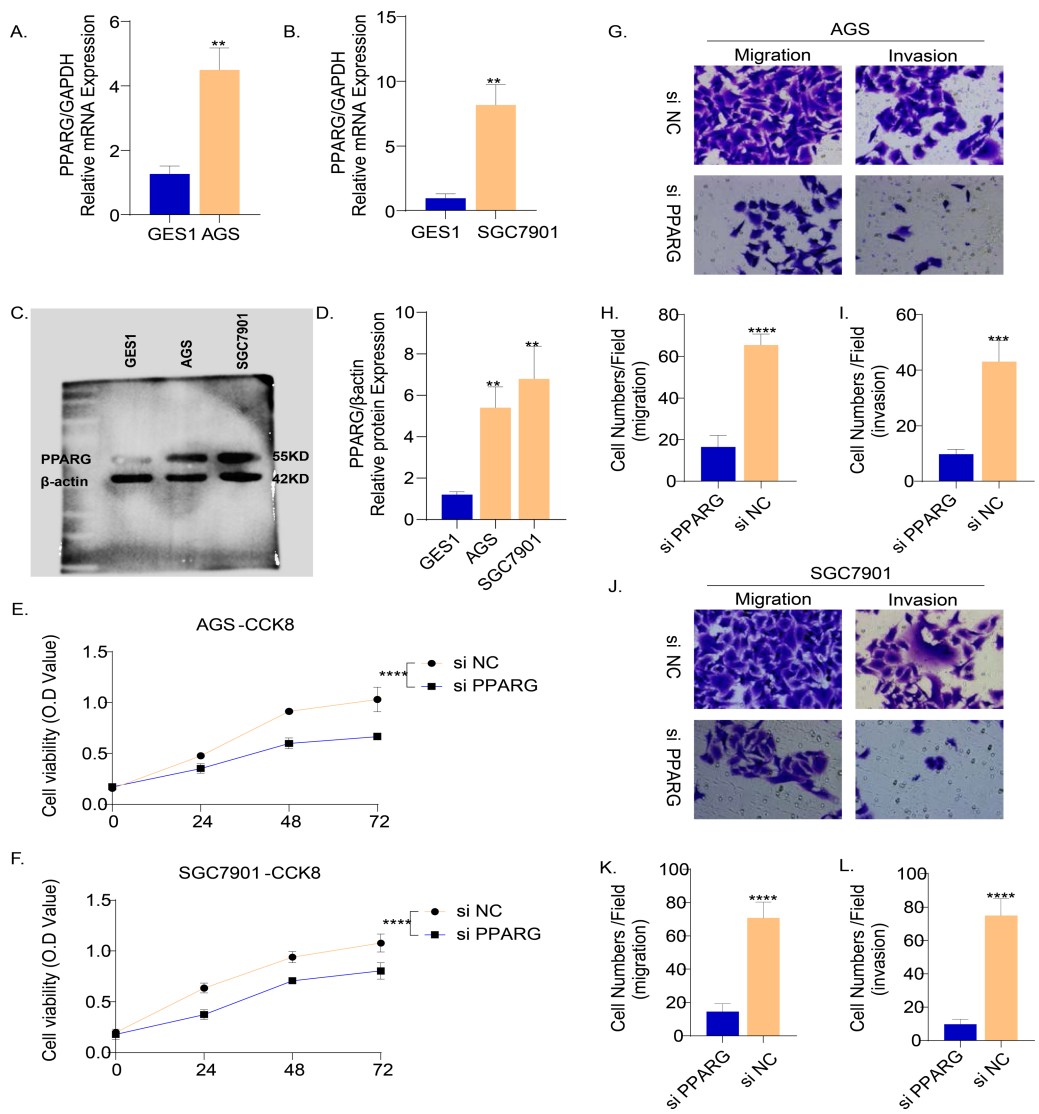

**Figure 9** **PPARG promotes the invasion and migration ability of gastric cancer cell lines.** (A–B) Expression of PPARG mRNA in three cell lines, GES1, AGS and SGC7901. (C–D) Expression of PPARG protein level in three cell lines GES1, AGS, SGC7901. (E–F) Cell viability of AGS and SGC7901 after PPARG knockdown. (G) Migration and invasive ability of AGS after PPARGA knockdown. (H–I) Quantitative statistics for cell counts. (J) Migration and invasive ability of SCG7901 after PPARGA knockdown. (K–L) Quantitative statistics for cell counts. $**p \leq 0.01$, $***p \leq 0.001$, $****p \leq 0.0001$. The results are presented as mean ±SD. $n = 3$/group.

ASAP2 in GC cell promotes cancer progression through regulating miR-770-5p level, while knockdown of ASAP2 can suppress cisplatin (DDP) resistance in GC treatment (*Sun et al., 2021*). Overexpressed DNM2 often leads to poor prognosis and enhanced invasive phenotype in many cancers (*Trochet & Bitoun, 2021*). KIF13B interacts with the tubulin VEGFR2, and KIF13B depletion prevents VEGF-mediated neo-vascularization and endothelial migration(*Yamada et al., 2014*). EAF2 is a potential tumor suppressor that

interacts with the eleven-nineteen lysine-rich leukemia (ELL) protein to inhibit cancer progression of B-cell lymphoma, prostatic cancer and hepatocellular carcinoma(*Pascal et al., 2019*; *Xiao et al., 2008*). MFSD9 encodes a transport protein, but its function in cancer has not been reported. Autophagy mediated by transmembrane protein TMEM164 degrades ferritin to promote ferroptosis (*Liu et al., 2022*; *Liu et al., 2023a*). These genes were closely related to PPARs and they all have critical functions in cancer progression, indicating the complex roles of PPARs in STAD.

The progression and invasion of GC are closely related to the biological characteristics of GC environment (*Rojas et al., 2020*). Hypoxic microenvironment, acidosis microenvironment and immune inflammatory response are also involved in the development of GC (*Peinado, Lavotshkin & Lyden, 2011*; *Schlappack, Zimmermann & Hill, 1991*; *Yang, Meng & Wang, 2021*). In our research, the PPAR genes were closely associated with T cell CD4 memory resting, NK cell resting, macrophages M0 and immune checkpoints genes such as PD-1 and PDL1. CD4+ T cell fraction is increased in GC (*Lu et al., 2017*). PPAR-$\gamma$ inhibition stimulates the differentiation of M2c macrophage-like (CD206(+) CD163(+) CD16(+)) cells (*Zizzo & Cohen, 2015*). Another study confirmed that PPAR-induced fatty acid oxidation in T cells can improve the efficacy of anti-PD-1 therapy (*Chowdhury et al., 2018*), indicating that the expression of PPARs is crucial for immune activation and patients' benefit from immune treatment.

TMB is associated with enhanced clinical immunotherapy response in STAD. *Samstein et al. (2019)* analyzed the correlation between TMB and OS of cancer patients treated with immune checkpoint inhibitor therapy, and they found a positive relationship between higher somatic TMB (highest 20% in each histology) and better OS in a variety of cancer types. Previous studies also reported that MUC16 mutation is predictive of a better prognosis in GC (*Jia et al., 2019*; *Li et al., 2018*). Mutation frequencies of MUC16 and TTN are closely correlated with TMB in GC (*Yang et al., 2020*). In this study, TMB in STAD was also closely related to PPARA and PPARG and the expression of TTN, MUC16, PPARA, and PPARG was positively related. However, despite interesting findings, the current study still had several limitations. Firstly, all the datasets were public databases and the sample size was relatively small. Secondly, the reliability of our findings should be further validated *in vivo* and *in vitro*. In addition, several factors including region differences and the presence of other disease should be considered for further analysis.

## CONCLUSION

In summary, this study was the first to comprehensively analyze the expression patterns of PPARA, PPARD and PPARG in multiple cancers including in STAD. The three PPAR genes were closely related to the prognosis, immune microenvironment, and genome mutation of STAD.

### Funding

The authors received no funding for this work.

## Competing Interests

The authors declare that there are no competing interests.

## Author Contributions

- Qing Jia conceived and designed the experiments, analyzed the data, authored or reviewed drafts of the article, and approved the final draft.
- Baozhen Li conceived and designed the experiments, prepared figures and/or tables, authored or reviewed drafts of the article, and approved the final draft.
- Xiulian Wang conceived and designed the experiments, analyzed the data, prepared figures and/or tables, and approved the final draft.
- Yongfen Ma performed the experiments, authored or reviewed drafts of the article, and approved the final draft.
- Gaozhong Li performed the experiments, analyzed the data, prepared figures and/or tables, authored or reviewed drafts of the article, and approved the final draft.

## Data Deposition

The datasets generated and/or analyzed during the current study are available in the GSE repository.

The raw data is available at GitHub:

- https://github.com/1Ligaozhong/Source-data.git

- 1Ligaozhong. (2023). 1Ligaozhong/Source-data: First release of my source data (v.1.0.0). Zenodo. https://doi.org/10.5281/zenodo.10076985.

## Supplemental Information

Supplemental information for this article can be found online at http://dx.doi.org/10.7717/peerj.17082#supplemental-information.

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
