# Peer review of "Comprehensive analysis of peroxisome proliferator-activated receptors to predict the drug resistance, immune microenvironment, and prognosis in stomach adenocarcinomas"

_PeerJ, doi:10.7717/peerj.17082_

## Round 0.1 · original submission · Major Revisions

Both reviewers provided many comments and suggestions on your manuscript. Please respond to these comments one by one.

**Language Note:** PeerJ staff have identified that the English language needs to be improved. When you prepare your next revision, please either (i) have a colleague who is proficient in English and familiar with the subject matter review your manuscript, or (ii) contact a professional editing service to review your manuscript. PeerJ can provide language editing services - you can contact us at [email protected] for pricing (be sure to provide your manuscript number and title). – PeerJ Staff

Reviewer 1 ·

Basic reporting

This study explored the relationship between PPARs and immune status, molecular mutations and drug therapy in gastric adenocarcinoma samples. It was confirmed that PPARs genes were closely associated with STAD, which suggested that PPARs genes might mediate themselves to STAD through immune and gene mutation pathways. The results were verified by cellular assays, which provided a new perspective to explore the molecular mechanism of gastric cancer. There are some issues in the manuscript, where the following comments were carefully answered.
1. In the manuscript, stomach adenocarcinomas and gastric cancer are used, which should be standardized.
2. In the abstract, when ssGSEA appears for the first time, the full name should be stated. ssGSEA is Single-sample gene set enrichment analysis, and in line 38, it should be ssGSEA results and not ssGSEA analysis.
3. In the second paragraph of the introduction, the research of PPARs in cancer is too old. It should introduce the research progress of PPARs in gastric cancer and the significance of research in gastric cancer.
4.The databases used in the methods, TCGA dataset, CCLE dataset, GDC dataset, GDSC dataset, should provide their URLs. At the end of Materials and Methods, it is necessary to add the methods and software used for statistical analysis and the judgment conditions of statistical significance.
5、Materials and methods in some R package or analysis method are missing references, ggplot2 package, CIBERSORT method, ssGSEA analysis, heatmap.

Experimental design

no comment

Validity of the findings

no comment

Additional comments

1, In the drug sensitivity analysis, in the absence of the validation of the cell experiment, is it reasonable to use only IC50 data as a predictor.
2, In lines 199-200, six genes were identified as highly associated with PPARs genes. what is the association of ASAP2, DNM2, EAF1, KIF13B, MFSD9, TMEM164 with STAD. What is their significance in STAD, it is suggested to add relevant discussion in the Discussion.
3, BP analysis, CC category, MF category in line 203 all belong to Gene ontology analysis. there is no full name of BP, CC, MF in the method, which is easy to cause trouble to non-specialists. This part should be explained in detail in the method.
4, The gene sets used in lines 210-211 are not mentioned in the Materials and Methods and should be added. The URL of the GSEA website should also be provided.
5. In the manuscript, regarding the results of correlations, such as Figure 6A, it is recommended that more data (for example, p-values, specific values of correlation coefficients, and criteria for statistical significance) be provided.
6, Regarding the results of Figure 6, a detailed discussion of the results is not mentioned in the manuscript. In the discussion section, it is acceptable for the authors to explain the biological significance of the findings, the consistency or inconsistency with previous studies, and to discuss possible mechanisms or clinical applications.
7, In some studies, it has been found that cancer patients with high TMB may respond better to immunotherapy. Regarding the results of Figure 7, if the analysis of IPS and TIDE scores in the PPARs group is increased, combined with the results of TMB, the clinical significance of PPARs in gastric cancer can be better illustrated.
8, All Figures definition are low-quality, it is need to be improved.
9, What does *, **, ***, ****, ns in Figure 1, Figure 5, Figure 6, Figure 7B stand for.

Reviewer 2 ·

Basic reporting

no comment

Experimental design

no comment

Validity of the findings

no comment

Additional comments

The subject of this study is to construct assessment models based on PPARs that can predict drug resistance, immune microenvironment characteristics and prognosis of gastric adenocarcinoma (STAD), and to validate their reliability by cellular experiments. This study first downloads PPARs expression profiles from public databases and reveals the relationship between PPARs and clinical features, prognosis, tumor microenvironment, genomic mutations, and drug sensitivity of STAD through a series of bioinformatics tools. Follow-up studies clarified the aberrant expression of PPARs in STAD cell lines through cellular experiments and the regulatory role of this expression on the malignant phenotype of cancer cells. In conclusion, the overall idea of the study is moderate and the experiments are reproducible, but the following issues still need to be addressed before publication:
1. The background section of the article clearly describes the relationship between PPARs and STAD. It is recommended that the introduction section elaborate in more detail on the role of PPARs in the occurrence and development of STAD and why this particular area was chosen for study.
2. In the discussion section, the authors need to explore in more depth the function and mechanism of PPARs genes and their potential therapeutic applications in STAD. A comparison with existing literature is suggested to discuss the novelty and limitations of this study.
3. Improved treatments for gastric cancer are mentioned in the Introduction, and it is suggested to expand on whether there are new treatments that can improve the prognosis of gastric cancer that are different from traditional treatments, and it is suggested to highlight the shortcomings of the existing medications for treating gastric cancer.
4. The main function of PPARs should be to regulate hormone secretion and cellular energy metabolism, and it is suggested that a detailed description of how cellular metabolism or hormone secretion processes regulated by PPARs act on cancer progression should be provided in the Introduction section, with particular emphasis on the mechanisms by which they act to regulate the malignant phenotype of gastric adenocarcinoma cells.
5. The Introduction section on the significance of the research in this paper is not sufficient and does not highlight the importance of bioinformatic tools in STAD research, please supplement the relevant literature to illustrate the current studies related to the effective mining of cancer therapeutic targets and explain how these studies are relevant to the development of this paper.
6. Figure 1 examines the expression of PPARs in pan-cancer, why was this step of the study done? Does this step of the study provide a reference for the direction of the later study? However, the results of the study did not show the specificity of PPARs in STAD, please give a reasonable explanation.
7. Figure 3 KM survival analysis of various types of PPARs demonstrated the association of PPARs with patient prognosis, is it possible to construct risk score models that quantify the expression of these PPARs to predict the prognosis of STAD through regression analysis?
8. What do the results of the "Pathways enrichment analysis" section tell us, and what implications do they have for later studies? In addition, the results are only stacked with functional differences in PPARGs between samples, what are the implications of these differences for STAD progression or drug resistance? Please describe systematically and add summary statements.
9. The association between PPARs and cancer progression has been demonstrated in the text, but whether the gene affects STAD progression through mechanisms such as tumor immune regulation, metabolic regulation, and gene mutation has not been mentioned in the Discussion section, so please add a relevant statement.
10. According to material dialectics, the research in this paper is bound to have certain flaws in addition to the content that deserves to be recognized, so please systematically express the limitations of this paper and add what subsequent means or experiments are proposed to be used to improve this research.

---

## Round 0.2 · accepted · Accept

Based on the opinions of the 2 reviewers, your manuscript is acceptable in principle. During the proofreading stage, please provide images with a resolution that meets the publication requirements of the magazine. In order to allow readers to better understand the volcano diagram in Fig 4, the gene names in the diagram can be retained.

Reviewer 1 ·

Basic reporting

Look at the later report

Experimental design

Look at the later report

Validity of the findings

Look at the later report

Additional comments

The manuscript focuses on the role of peroxisome proliferator-activated receptors (PPARs) in stomach adenocarcinomas (STAD), analyzing their impact on drug resistance, the immune microenvironment, and patient prognosis. Utilizing data from TCGA and other databases, the study examines the expression patterns of PPAR genes and their associations with clinical features, immune status, and drug sensitivity. The findings reveal that PPARs significantly influence STAD's molecular landscape, suggesting that targeting these pathways could offer new therapeutic strategies. I find the study provides valuable insights into the complex role of PPARs in STAD, supported by comprehensive data analysis. However, the manuscript would benefit from further validation of the findings through experimental approaches, considering the potential impact of regional differences and comorbidities. Additionally, addressing the relatively small sample size and enhancing the clarity of data presentation could strengthen the manuscript's contribution to the field.

Reviewer 3 ·

Basic reporting

no comment

Experimental design

no comment

Validity of the findings

no comment

Additional comments

I basically agree with the comments of the previous two reviewers, and the author has provided detailed responses. Overall, the manuscript has been greatly improved. I don't have many comments, and the quality of the images needs to be improved. Is it necessary to display gene names in the volcano diagram in Fig4? If not, it is recommended to delete them before publication.